# Early Hemorrhagic Complications after Holmium Laser Enucleation of the Prostate in Patients Undergoing Antithrombotic Therapy: A Retrospective Analysis from a High-Volume Centre

**DOI:** 10.3390/jcm13196006

**Published:** 2024-10-09

**Authors:** Serena Pastore, Marco Carilli, Stefano Di Nicola, Adriano Campagna, Ulderico Parente, Federico Pierella, Giulia D’Ippolito, Enrico Finazzi Agrò, Alessio Zuccalà

**Affiliations:** 1Urology Unit, Aurelia Hospital, Via Aurelia 860, 00165 Rome, Italy; sere.pastore@gmail.com (S.P.);; 2Urology Unit, San Carlo di Nancy Hospital–GVM Care and Research, Via Aurelia 275, 00165 Rome, Italy; 3Department of Surgical Sciences, Tor Vergata University of Rome, Via Montpellier 1, 00133 Rome, Italy; finazzi.agro@med.uniroma2.it

**Keywords:** HoLEP, anticoagulants, complications, bleeding, lower urinary tract symptoms

## Abstract

**Objectives:** This study intends to evaluate early hemorrhagic complications after holmium laser enucleation of the prostate (HoLEP) in patients undergoing antithrombotic therapy. **Methods:** The data of patients undergoing HoLEP between January 2020 and February 2023 were retrospectively analysed. Patients were clustered into three groups: (1) no antithrombotic therapy; (2) antiplatelet (AP) therapy; and (3) anticoagulant (AC) therapy. Pre-, intra-, and post-operative variables were compared. A logistic regression model was built to identify predictors of post-operative hemorrhagic complications. **Results:** A total of 338 patients underwent HoLEP, including 212 who received no antithrombotic therapy (62.7%), 76 who received AP (22.5%), and 50 who received AC (14.8%). Intra-operative outcomes did not show any significant difference. A significant difference was observed in terms of catheterisation time (*p* = 0.001) and length of hospital stay (*p* < 0.001), favouring patients who did not receive antithrombotic therapy. Early post-operative hemorrhagic complications (<30 days) included re-admissions for macrohematuria (3.5%), transfusions (2.4%), and endoscopic re-interventions for bleeding (1.2%). A comparison between the groups showed significant differences for both re-admission (*p* < 0.001) and transfusion rates (*p* = 0.01), favouring patients who did not receive antithrombotic therapy. The re-intervention rate did not show any significant difference between the groups (*p* = 0.1). In multivariate analysis, AC therapy was identified as an independent predictor of those complications (OR 4.9, *p* = 0.005). **Conclusions:** HoLEP is a safe and effective procedure for patients undergoing antithrombotic therapy. Both AP and AC therapies are associated with longer catheterisation and hospitalisation times. AC therapy is shown to be a predictor of minor post-operative hemorrhagic complications.

## 1. Introduction

Holmium laser enucleation of the prostate (HoLEP) is a cornerstone in the surgical treatment of lower urinary tract symptoms (LUTSs) secondary to benign prostatic obstruction (BPO), since it allows practitioners to virtually manage any prostate volume [1]. For this reason, in high-volume centres with laser technology availability, HoLEP is gradually replacing the historic endoscopic gold standard, namely, transurethral resection of the prostate (TURP).

Another reason for the widespread adoption of this technique is due to its role in treating patients undergoing antithrombotic therapy. Indeed, antiplatelet (AP) and anticoagulant (AC) therapies are recognized as major risk factors for post-operative bleeding after TURP, which is a source of great concern for the urologist [2,3]. Since their introduction in clinical practice, lasers for BPH endoscopic surgery proved to be more efficient in terms of hemostatic control when compared to TURP [4]. HoLEP is not an exception, and its efficacy and safety has been proven even in patients undergoing antithrombotic therapy [5,6].

However, most of the available literature on this topic combined patients undergoing AP and AC therapies, and did not discriminate the results according to the antithrombotic drug [7]. Moreover, little is known about new oral anticoagulants (NOACs) and endoscopic laser procedures, despite those drugs gradually replacing dicumarolic agents as the standard of care since their introduction in 2010. They offer several advantages over traditional vitamin K antagonists, including lack of routine lab monitoring, fewer drug dietary interactions, shorter reversal time, and fewer major bleeding events. Nevertheless, their safety profile in patients scheduled for HoLEP is still unclear [8,9,10].

The aim of the present study was to evaluate early hemorrhagic complications after HoLEP in patients undergoing antithrombotic therapy, in order to better define the safety profile of HoLEP in this specific setting. 

## 2. Materials and Methods

### 2.1. Patients

The data of patients affected by BPO and LUTSs refractory to medical treatment or with intolerance to medical therapy with indication to surgical intervention (maximum flow rate (Qmax) < 15 mL/s and International Prostate Symptom Score (IPSS) >12) undergoing HoLEP between January 2020 and February 2023 were included in this retrospective analysis.

The study was conducted in accordance with the Declaration of Helsinki and its later amendments. The study was approved by the local institutional ethics committee and informed consent was taken from all the patients.

Exclusion criteria were as follows: previous prostate surgery; urethral strictures; bladder stones; a history of prostate or bladder cancer; neurogenic bladder dysfunction; or urinary incontinence.

For the purpose of the study, patients were clustered into three groups: (1) no antithrombotic therapy; (2) AP therapy; and (3) AC therapy.

### 2.2. Data Collection and Outcome Measurements

Pre-operative variables, including age, the American Society of Anesthesiologists (ASA) score [11], the serum prostate-specific antigen (PSA), the prostate volume (assessed by transrectal ultrasound), the presence of an indwelling catheter (yes/no), hemoglobin (Hb), and the maximum flow rate (Qmax) with post-voiding residual volume (PVR) were collected at baseline.

Intra- and perioperative data were collected, including the operative time, the enucleation volume, the velocity of tissue retrieval (i.e., the ratio between operative time and enucleation volume), the length of hospital stay, the catheterisation time, and drop in Hb on the 1st post-operative day.

Eventual early post-operative hemorrhagic complications (within 30 days) were recorded. Those included (1) re-admissions for persistent macrohematuria managed with observation or bladder irrigation; (2) blood transfusions; and (3) endoscopic re-interventions for active bleeding refractory to conservative management.

Finally, post-operative functional outcomes were evaluated at 30 days by uroflowmetry (Qmax and PVR).

### 2.3. Surgical Procedure and Post-Operative Protocol

All procedures were performed in a referral endourological institution (Urology Unit, Aurelia Hospital, Rome, Italy) under general anesthesia with a continuous-flow 26 Ch resectoscope (Karl Storz (Tuttlingen, Germany)) and a 550 μm holmium laser end-fire fiber connected to the Lumenis Pulse^TM^ 120H generator. The procedures were performed using a power setting of 80–100 W with 2 J and 50 Hz, without any modifications during coagulation or apical dissection; generally, a short-pulse duration was used during enucleation. HoLEP was performed with a classic “three-lobe” or modified “two-lobe” technique [12], depending on the prostate anatomy. A rigid nephroscope with a 5 mm working channel was used to introduce the tissue morcellator at the end of the procedure using a dual irrigation system. A non-toothed guillotine blade morcellator (VersaCut^TM^, Lumenis (Yokneam, Israel)) was used to remove the enucleated tissue.

The post-operative standard protocol was made up of the following: (1) stopping bladder irrigation on the 1st post-operative day; (2) catheter removal on the 2nd post-operative day; and (3) discharge after spontaneous micturition. As stated above, any deviation from this protocol was recorded.

### 2.4. Perioperative Management of Antithrombotic Therapy

As stated in the European Association of Urology (EAU) guidelines [13], patients undergoing AP therapy discontinued it 5–7 days before intervention, and then re-started it on the 5th post-operative day. In detail, acetylsalicylic acid was stopped only in the case of primary thromboprophylaxis, while patients with a history of acute myocardial infarction or cerebrovascular disease continued their therapy throughout surgery. Other AP drugs (e.g., clopidogrel) were discontinued 7 days before HoLEP.

Regarding AC therapy, dicumarolic agents (e.g., warfarin) were discontinued 7 days before surgery, and patients were prescribed bridging therapy with low-molecular-weight heparins (LMWHs) up to the 7th post-operative day. Conversely, patients intaking direct thrombin inhibitor (dabigatran) or factor Xa inhibitors (apixaban, edoxaban, rivaroxaban) discontinued them 48 hours before HoLEP without any bridging therapy, and re-started NOAC therapy 48–72 hours after surgery.

### 2.5. Statistical Analysis

Continuous variables were summarized using medians and interquartile ranges (IQRs); frequencies and proportions were used to report categorical variables. All data were tested for normality using the Shapiro–Wilk test. The Kruskal–Wallis and Mann–Whitney non-parametric tests were performed to compare continuous variables among the groups. A Chi-square test was used for categorical variables. A logistic regression model was built to identify predictors of post-operative hemorrhagic complications. Data analysis was conducted using SPSS 21.0 software (IBM, Armonk, NY, USA). Statistical significance was defined as a *p*-value < 0.05.

## 3. Results

A total of 338 patients who underwent HoLEP were eligible. A total of 212 patients (62.7%) did not receive antithrombotic therapy, 76 (22.5%) were treated with AP therapy, and 50 (14.8%) were treated with AC therapy. Among the AP cohort, 37 patients (48.7%) received acetylsalicylic acid, while 39 (51.3%) received clopidogrel. Regarding the AC cohort, 3 patients (6%) received warfarin, while 47 (94%) received NOACs (38 direct thrombin inhibitor and 9 factor Xa inhibitors, respectively). Table 1 shows the distribution of baseline patient characteristics: groups were comparable at baseline, with statistically significant differences observed for age (*p* < 0.001) and ASA score (*p* = 0.01).

In Table 2, peri- and post-operative outcomes are reported. Intra-operative outcomes did not show any significant difference. Regarding the early post-operative course (within 7 days), a significant difference was observed in terms of the catheterisation time (*p* = 0.001) and the length of hospital stay (*p* < 0.001), favouring patients who did not receive antithrombotic therapy. A post hoc analysis showed no difference between the AP and AC therapies for those variables (both *p* = 0.3). Moreover, a statistically significant difference between groups was observed regarding the number of patients deviating from the standard post-operative protocol because of macrohematuria (*p* = 0.001).

Overall, post-operative hemorrhagic complications (within 30 days) included 12 re-admissions for persistent macrohematuria (3.5%), 8 transfusions (2.4%), and 4 endoscopic re-interventions for bleeding (1.2%). A comparison between the groups showed significant differences for both re-admission (*p* < 0.001) and transfusion rates (*p* = 0.01), favouring patients who did not receive antithrombotic therapy. A post hoc analysis showed a significant difference between AP and AC therapies for re-admission rate (2% vs. 16%, *p* = 0.007), while this difference was not observed in transfusion rate (2.6% vs. 8%, *p* = 0.2). The re-intervention rate did not show any significant difference between groups (*p* = 0.1).

Post-operative functional outcomes were comparable among the groups, as witnessed by the raw point improvement in Qmax and PVR, respectively.

As shown in Table 3, the univariate logistic regression model identified age (odds ratio [OR] 1.1, *p* = 0.03), ASA score III (OR 3.5, *p* = 0.02), and AC therapy (OR 6.7, *p* < 0.001) as predictors of minor post-operative hemorrhagic complications (i.e., re-admissions or transfusions). In multivariate analysis, only AC therapy was confirmed as an independent predictor of those complications (OR 4.9, *p* = 0.005).

## 4. Discussion

As shown in the latest EAU guidelines, HoLEP currently represents the first choice for treatment of LUTSs secondary to BPO, with indication to surgical intervention and prostate volume > 80 mL, given the more favourable peri-operative profile compared to open prostatectomy [1]. On the other hand, in prostate volumes between 30 and 80 mL, transurethral resection of the prostate (TURP) is still considered the gold standard; however, HoLEP can be considered a valid alternative to TURP in experienced hands, since several meta-analyses showed that HoLEP is characterized by longer operative times, but shorter catheterisation and hospitalisation times, reduced blood loss, and fewer blood transfusions [14,15].

Historically, a major concern about prostatic endoscopic procedures has been bleeding prevention and management. These issues are even more complicated in patients undergoing antithrombotic therapy, in which the risk of bleeding complications and blood transfusions should be carefully taken into account. As is well known, the incidence of chronic aging-associated diseases, especially cardiovascular and prostatic diseases, is increasing with the aging of society [16]. Nowadays, urologists have to deal with elderly patients with multiple comorbidities, and who are possibly undergoing AP or AC therapies; consequently, searching for surgical techniques that best fit with these issues is critical. Due to its physical features (pulsed, solid-state 2140 nm wavelength, tissue penetration of about 0.4 mm, coagulation of small- and medium-sized vessels to a depth of 2–3 mm), a holmium laser allows for rapid hemostasis, which is beneficial when operating on patients taking AP or AC drugs [17].

Bearing in mind the above-cited considerations, the present study was conceived to evaluate early hemorrhagic complications (within 30 days) after HoLEP in patients undergoing antithrombotic therapy, in order to better define its safety profile in this specific setting.

Results from this study showed that 126 patients (37.3% of cases) scheduled for HoLEP received antithrombotic therapy prior to surgery. At baseline, these patients were older and with higher ASA scores compared to patients taking no therapy. Antithrombotic therapy was not associated with worse intra-operative outcomes, though significant differences were found in catheterisation and hospitalisation times, as well as in minor post-operative hemorrhagic complications (i.e., re-admissions and transfusions). The study’s cohorts did not show any differences in post-operative functional outcomes. Finally, AC therapy was identified as an independent predictor of minor post-operative hemorrhagic complications, as shown by the logistic regression model.

To date, no prospective study nor randomized clinical trial has been published on this topic; consequently, the available evidence comes exclusively from retrospective studies, and results are rather heterogeneous. The first report on HoLEP and antithrombotic therapy was published in 2002 by Hochreiter et al., which reported treatment of 19 patients while under complete oral AC and a median international normalized ratio of 2.7: no patient required blood transfusion, although two patients (10.5%) developed clot retention, which was managed conservatively [18].

Since this initial experience, several HoLEP studies on patients taking antithrombotic drugs have been published. In 2006, Elzayat et al. reported a series of 81 patients on antithrombotic therapy: 14 patients continued AC/AP throughout surgery, 34 underwent bridging with LMWH, and 33 stopped AC/AP before surgery. Blood transfusions were required in eight patients (9.6% of cases); within the group of patients requiring a blood transfusion, two patients were on full AC, five patients were undergoing bridging therapy, and one patient was coming off of AC/AP therapy. There were no major post-operative complications [19].

In another study, Tyson and Lerner compared 39 patients on antithrombotic therapy (warfarin or aspirin) with 37 controls undergoing HoLEP. No statistically significant difference was noted between the groups in terms of hemorrhagic complications (e.g., blood transfusion rate, re-admissions for haematuria, or continuous bladder irrigation) [20]. The major limitation of this study, which inevitably affects the interpretation of its results, is the small sample size.

Bishop et al. compared 52 patients on antithrombotic therapy at the time of HoLEP (aspirin 100–150 mg daily, clopidogrel 75 mg daily, dipyridamole 200 mg/aspirin 25 mg twice daily, or warfarin) with 73 patients not on therapy (including those who stopped antithrombotic agents for an adequate period of time prior to surgery). Patients undergoing antithrombotic therapy were older, had a higher ASA score at baseline, and had a longer hospitalisation time. The transfusion rate (7.7%) was higher in the antithrombotic cohort, although no patients required re-operation for bleeding [21].

Another study by El Tayeb et al. compared 116 patients on AC/AP therapy with 1558 patients not on therapy; patients were also divided into continuous vs. intermittent AC/AP therapy (i.e., patients who discontinued the therapy pre-operatively and re-started it 1 week post-operatively) and compared with each other. Interestingly, this was the first report including NOACs (specifically, dabigatran). Results from this study showed no pre-, intra-, or post-operative differences among the groups [22].

More recently, Becker et al. focused their attention exclusively on patients taking AC therapy (namely, 94 patients receiving NOACs and 151 receiving warfarin), but excluding patients on AP therapy. Patients stopped NOACs 48 hours before HoLEP, while warfarin was stopped 10 days before surgery; all patients were bridged with LMWH for 14 days before re-starting the treatment. This cohort of AC patients was compared with 1933 non-anticoagulated patients. Patients on AC therapy showed a higher risk of intra- and post-operative bleeding complications (bladder tamponades 6.5%, re-intervention for endoscopic coagulation 3.7%, and blood transfusions 1.6%), although still significantly lower compared to TURP [23].

In 2020, Boeri et al. performed a propensity-score matching between patients who underwent HoLEP on AC or AP therapy, or no therapy. Of note, this series did not include patients on NOACs. A total of 28 and 46 patients had AC and AP therapy, respectively. HoLEP patients under either AC or AP therapy required longer catheterisation and hospitalisation times than those without AC/AP therapy. Operative time, post-operative complications, and symptom scores were similar between groups [24].

Evaluating the safety profile of 104 patients in antithrombotic therapy undergoing HoLEP, Deuker et al. analysed functional outcomes and early post-operative complications, stratified according to the specific drug (platelet aggregation inhibitors (or PAIs), NOACs, antithrombotic combination, or coumarins bridged with LWMH). In this report, 8.2% of patients received NOACs, while 4.9% received an LWMH/combination. Significant differences were recorded for overall early complication rates (26.1% vs. 27.3% vs. 46.2%, respectively, in PAI, NOACs, and LMWH/combination patients). Minor complications were significantly more frequent in patients with antithrombotic therapy, while no difference was noted in major complications. In the multivariate logistic regression model, antithrombotic therapy was not significantly associated with higher complication rates, whereas a high ASA score (OR 2.2, *p* = 0.04), age (OR 1.04, *p* = 0.02), and bioptical or incidental prostate cancer (OR 2.5, *p* = 0.01) represented independent risk factors [25].

Focusing on the role of NOAC therapy before HoLEP, Agarwal et al. reported the outcomes of 65 patients on AC therapy, including 42 (64.7%) patients taking NOACs and the remaining 23 (35.3%) taking warfarin. Comparing these drugs, warfarin showed a lower rate of successful catheter removal, a higher 90-day complication rate, and more re-admissions to the Emergency Department compared to NOACs [26].

Finally, a recent review by Netsch et al. reported the highest transfusion rates in patients with LMWH bridging or under continuous AC therapy (up to 15%), while transfusion rates under AP therapy did not exceed 3%; post-operative clot retention occurred in a maximum of 12.5% of the patients, while re-intervention rates were reported in a maximum of 3.7% of the cases [6]. These percentages are comparable to our findings.

Our results require some interpretation. As expected, patients undergoing AP/AC therapies are older and more comorbid. Intra-operative outcomes and drop in Hb on the 1st post-operative day were similar between groups, while the statistically significant differences in peri- and post-operative outcomes are visible starting from the 2nd post-operative day, when the antithrombotic therapy (in particular, NOACs) is usually restored after HoLEP. This observation is consistent with NOAC pharmacokinetics, with time to maximum plasma concentration (t_max_) ranging from 1 to 4 hours and terminal half-life (t_1/2β_) ranging from 12 to 17 hours [27]. Moreover, consistent with previous reports, antithrombotic therapy is associated mostly with minor hemorrhagic complications after transurethral prostatic procedures, even if stopped pre-operatively: probably, these complications are not related to drug-induced spontaneous haemorrhages, but rather to an interference with the mechanism of healing of the prostatic urethra following thermal injury (the so-called “eschar fall”, which generally occurs 15–20 days after surgery) [28].

As stated above, a statistically significant difference in terms of catheterisation and hospitalisation times was observed between groups, disfavouring antithrombotic therapy. In our opinion, this can represent a significant limitation for same-day catheter removal and hospital discharge protocols following HoLEP. Indeed, this is not surprising, as a recent retrospective analysis by Badreddine et al. showed that a history of AP/AC use was predictive of same-day discharge failure [29].

We acknowledge the limitations of this study. First, we acknowledge the retrospective nature of this study. Second, the small sample size does not allow us to draw any definitive conclusion, especially in distinguishing between warfarin’s and NOACs’ outcomes. However, to the best of our knowledge, this is one of the largest reports on NOACs before HoLEP (94% of AC cohort). This high percentage is probably related to the fact that Aurelia Hospital is also a referral centre for Interventional Cardiology and Cardiovascular Surgery; consequently, the percentage reflects this aspect, with more patients under antithrombotic therapy referring for BPH surgery. Third, our results could be influenced by some biases, including the number of the surgeons involved (although all of them had consolidated experience in HoLEP) and the lack of a full pharmacological anamnesis (indeed, drug interaction may have increased the risk of bleeding in some patients). Regarding this latter issue, NOAC drug–drug interactions in patients without renal impairment are not fully understood: as reported in a recent review, instead of avoiding drug combinations with NOACs, more trials should be conducted and new strategies such as dose adjustments based on therapeutic drug monitoring should be investigated [30]. Finally, despite it not being the main objective of our study, post-operative functional assessment relied only on uroflowmetry parameters, without any data about patient-reported outcomes.

Notwithstanding these limitations, the results of this study confirm the safety profile of antithrombotic therapy in patients scheduled for HoLEP. Although our study does not allow us to establish any strong conclusions, we underline the paramount importance of adequate pre-operative counselling in this specific setting.

## 5. Conclusions

HoLEP is a safe and effective procedure in patients undergoing antithrombotic therapy. Both AP and AC therapies are associated with longer catheterisation and hospitalisation times. Pre-operative counselling about the higher risk of minor post-operative complications is paramount. 

## Figures and Tables

**Table 1 jcm-13-06006-t001:** Baseline characteristics.

	Overall n = 338 (%)	No Therapy n = 212 (62.7)	AP Therapy n = 76 (22.5)	AC Therapy n = 50 (14.8)	*p*-Value
Age (years)	68 (62–74)	66 (62–72)	71 (68–76)	73 (64–77)	<0.001
ASA score					
ASA I	12 (3.5)	7 (3.3)	3 (3.9)	2 (4.0)	0.01
ASA II	273 (80.8)	180 (84.9)	61 (80.3)	32 (64.0)	
ASA III	53 (15.7)	25 (11.8)	12 (15.8)	16 (32.0)	
Prostate volume (mL)	75 (60–94)	75 (60–91)	70 (60–86)	75 (70–95)	0.09
PSA (ng/mL)	4.1 (2.4–6.7)	4.3 (2.4–6.9)	3.6 (1.8–5.7)	4.5 (2.7–7.2)	0.1
Indwelling catheter					
No	300 (88.8)	190 (89.6)	67 (88.2)	43 (86.0)	0.8
Yes	38 (11.2)	22 (10.4)	9 (11.8)	7 (14.0)	

Median and interquartile range (IQR) are reported for continuous variables, while numbers of observations with percentages (%) are reported for categorical variables. AP: antiplatelet, AC: anticoagulant, ASA: American Society of Anesthesiologists, PSA: prostate-specific antigen.

**Table 2 jcm-13-06006-t002:** Peri- and post-operative outcomes.

	Overall n = 338 (%)	No Therapy n = 212 (62.7)	AP Therapy n = 76 (22.5)	AC Therapy n = 50 (14.8)	*p*-Value
Intra-operative outcomes					
Operative time (min)	70 (55–90)	65 (50–90)	70 (55–96)	72 (60–82)	0.3
Enucleation volume (g)	43 (30–64)	45 (30–65)	41 (27–61)	46 (38–60)	0.3
Velocity of tissue retrieval (g/min)	0.6 (0.4–0.9)	0.7 (0.4–0.9)	0.6 (0.4–0.9)	0.7 (0.4–0.8)	0.2
Early post-operative course (≤1 wk)					
Catheter time (days)	2 (2–2)	2 (2–2)	2 (2–2)	2 (2–2)	0.001
LOS (days)	3 (3–3)	3 (3–3)	3 (3–3)	3 (3–3)	<0.001
Deviations from standard protocol	42 (12.4)	16 (7.5)	14 (18.4)	12 (24.0)	0.001
1st POD ΔHb (g/dL)	–2.1 (–2.3–−1.7)	–2.1 (–2.3–−1.8)	–2.0 (–2.2–−1.6)	–2.0 (–2.4–−1.6)	0.6
Post-operative complications (≤1 mo)					
Transfusion	8 (2.4)	2 (0.9)	2 (2.6)	4 (8.0)	0.01
Re-admission for macrohematuria	12 (3.5)	2 (0.9)	2 (2.6)	8 (16.0)	<0.001
Endoscopic revision for bleeding	4 (1.2)	1 (0.4)	1 (1.3)	2 (4.0)	0.1
Functional outcomes (1 mo)					
ΔQmax (mL/s)	+11.8 (+8.0–+15.8)	+12.5 (+8.6–+16.0)	+11.3 (+7.9–+15.0)	+ 9.0 (+ 5.8–+16.6)	0.2
ΔPVR (mL)	–80 (–150–0)	–80 (–126–0)	–100 (–200–0)	–90 (–120–0)	0.7

Median and interquartile range (IQR) are reported for continuous variables, while numbers of observations with percentages (%) are reported for categorical variables. AP: antiplatelet, AC: anticoagulant, LOS: length of hospital stay, POD: post-operative day, Qmax: maximum flow rate, PVR: post-void residual, PSA: prostate-specific antigen.

**Table 3 jcm-13-06006-t003:** Logistic regression model predicting minor hemorrhagic complications (≤1 mo).

	OR	95% C.I.	*p*-Value
Univariate analysis			
Age	1.1	1.0–1.2	0.03
ASA score III	3.5	1.2–10.1	0.02
AP therapy	1.2	0.4–3.7	0.8
AC therapy	6.7	2.4–18.7	<0.001
Prostate volume	1.0	0.9–1.0	0.6
Operative time	1.0	0.9–1.0	0.4
Enucleation volume	1.0	0.9–1.0	0.4
Multivariate analysis			
Age	1.0	0.9–1.1	0.2
ASA score III	2.0	0.6–6.4	0.2
AC therapy	4.9	1.6–14.7	0.005

OR: odds ratio, C.I.: confidence interval, AP: antiplatelet, AC: anticoagulant, ASA: American Society of Anesthesiologists.

## Data Availability

The raw data supporting the conclusions of this article will be made available by the authors on request.

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
