# Peer review of "Early Hemorrhagic Complications after Holmium Laser Enucleation of the Prostate in Patients Undergoing Antithrombotic Therapy: A Retrospective Analysis from a High-Volume Centre"

_jcm, 2024, doi:10.3390/jcm13196006_

Round 1

Reviewer 1 Report

Comments and Suggestions for Authors

This is an extremely interesting paper on a trending topic. Endoscopic enucleation of the prostate is tranforming into the new "gold standard" for BPH management and its potential benefit in patients receiving AC/AP therapy is of very high scientific interest. The authors must be complimented for the quality of their content and their proper methodology. The conclusions are well-supported by the results, while the Englifh expression is fine and the article can be read easily and pleasantly. Comments for the authors:

1) Were data about preoperative and postoperative IPSS and QoL available? If yes it would be interesting to presnet them in the tables and the manuscript of the paper.

2) Can the authors provide more information about the quipement (laser device, morcelator etc) and the energy settings that they used?

3) Are data about morcellation time available? It is highly recommend it to present the morcellation time along with the enucleation time.

4) The authors could discuss in the Discussion section of their study the perspective of same-day catheter removal and same-day hospital discharge following HoLEP and how AP/AC theramy may alter this approach.

5) Are there any practical solutions and tips and tricks available in the literarure regarding ways to improve outcomes in patients under AP/AC therapy (for example furosemide administration etc)? If yes, the authors could present them in the discussion section of their paper. 

Author Response

This is an extremely interesting paper on a trending topic. Endoscopic enucleation of the prostate is transforming into the new "gold standard" for BPH management and its potential benefit in patients receiving AC/AP therapy is of very high scientific interest. The authors must be complimented for the quality of their content and their proper methodology. The conclusions are well-supported by the results, while the English expression is fine and the article can be read easily and pleasantly. Comments for the authors:

Comments 1: Were data about preoperative and postoperative IPSS and QoL available? If yes it would be interesting to present them in the tables and the manuscript of the paper.

Response 1: Thanks for this point. Unfortunately, post-operative IPSS and QoL scores were available for a very limited number of patients, making any statistical analysis on patient-reported outcomes unreliable.  As stated among limitations, this was due to the retrospective nature of the study. However, we scheduled for HoLEP only patients with moderate-to-severe symptoms (pre-operative IPSS > 12), as described in Materials and Methods section.

Comments 2: Can the authors provide more information about the equipment (laser device, morcellator etc) and the energy settings that they used?

Response 2: Thanks for this suggestion. We included additional technical specifications in the “Surgical procedure” paragraph (pages 4-5, lines 106-112).

Comments 3: Are data about morcellation time available? It is highly recommend it to present the morcellation time along with the enucleation time.

Response 3: Thanks for this point. Due to the retrospective nature of the study, enucleation and morcellation times are not available. Nonetheless, we highlight that overall operative time, considered as “actual” surgical time (i.e. from the instrument introduction to the insertion of bladder catheter) showed no significant differences between groups (p = 0.3). In this sense, we believe that this variable alone is sufficient in demonstrating that antithrombotic therapy does not influence intra-operative outcomes.

Comments 4: The authors could discuss in the Discussion section of their study the perspective of same-day catheter removal and same-day hospital discharge following HoLEP and how AP/AC therapy may alter this approach.

Response 4: Very interesting suggestion. We added our perspective on this topic in Discussion section (page 10, lines 266-271) with a specific reference supporting our point of view.

Comments 5: Are there any practical solutions and tips and tricks available in the literature regarding ways to improve outcomes in patients under AP/AC therapy (for example furosemide administration etc)? If yes, the authors could present them in the discussion section of their paper. 

Response 5: The reviewer raised a very interesting question. To the best of our knowledge, there aren’t tips and tricks available in literature to improve post-operative outcomes of patients on antithrombotic therapy undergoing HoLEP. In our opinion, AP/AC therapy should be considered a non-modifiable risk factor for bleeding, even if managed as per EAU guidelines (see Materials and Methods section). Up to date, there are no evidences about the role of pulse modulation (for example, MOSES technology) in this specific setting: in 2022 a meta-analysis by Gauhar et al. comparing perioperative parameters of standard HoLEP vs MoLEP concluded that MoLEP performed better in terms of intraoperative outcomes (including intraoperative hemostasis time), but among the limitations of the study was included the lack of data about patients under AC therapy [doi: 10.1016/j.euf.2022.01.013]. This could represent an interesting field of research.

Reviewer 2 Report

Comments and Suggestions for Authors

The research paper entitled ``Early hemorragic complications after holmium laser enucleation of the prostate in patients undergoing antithrombotic therapy: a retrospective analysis from a high-volume centre`` is a retrospective study that evaluates early hemorragic complications after holmium laser enucleation of the prostate in patients undergoing antithrombotic therapy. There are few aspects that I like to mention:

- The retrospective nature or the study limits the ability to establish causality and may introduce selection bias. This design can lead to confounding variables that are not adequately controlled.

- Although the study claims to be among the larger reports regarding NOACs (New Oral Anticoagulants) before HoLEP, the overall sample size of 338 may still be inadequate to draw definitive conclusions, particularly when stratifying outcomes between specific antithrombotic medications (e.g., NOACs vs. warfarin).

- Lack of Distinction Between Antithrombotic Drugs: The study groups patients into broad categories (no therapy, AP therapy, AC therapy) without sufficiently analyzing the nuanced effects of different drugs within those categories. For instance, the outcomes of various NOACs or the effects of specific AP therapies were not evaluated in detail, which could affect the generalizability of the findings.

- The inclusion/exclusion criteria may introduce bias. For example, patients with a history of prostate or bladder cancer, which may relate to different risk profiles and complications, were excluded, potentially skewing results in favor of a more homogeneous patient population.

- Postoperative Outcomes Limited to Uroflowmetry: The evaluation of postoperative functional outcomes is limited solely to uroflowmetry parameters. This method does not capture holistic patient satisfaction or other relevant health-related quality-of-life indicators that could provide better insight into the procedure's overall efficacy.

- Inconsistent Preoperative Management: The management of antithrombotic therapy showed variability in application, particularly regarding bridging therapy. The inconsistency in management protocols across patients (e.g., varying timing for stopping and restarting therapies) could confound outcomes and complicate the interpretation of bleeding risk.

- Incomplete Pharmacological Anamnesis: The absence of a complete pharmacological history limits an understanding of potential drug interactions and their effects on bleeding risk, which has been highlighted as a critical area of concern for patients undergoing surgery while on antithrombotic therapy.

- No Randomized Control: The lack of a control group that includes a randomized setup could bias results. It would be beneficial to compare HoLEP outcomes in a controlled setup among patients with similar health backgrounds but differing in anticoagulant therapy type.

- Temporal Considerations: The study acknowledges that differences in outcomes appear postoperatively, around the second day, coinciding with the resumption of antithrombotic therapy. This temporal aspect may suggest that some outcomes could be mediated by factors linked to the timing of drug actions, which were not thoroughly examined.

Overall, while the study contributes valuable insights into the interaction between HoLEP and antithrombotic therapy, significant methodological limitations and biases must be addressed before considering these findings as definitive. Future research should aim for prospective, randomized controlled trials with larger and more diverse populations, comprehensive pharmacological assessments, and a broader range of outcome measures to enhance the reliability and applicability of the results. This could ultimately establish clearer guidelines for managing patients on antithrombotic therapy undergoing HoLEP, thereby improving clinical outcomes and patient care.

Author Response

The research paper entitled “Early hemorragic complications after holmium laser enucleation of the prostate in patients undergoing antithrombotic therapy: a retrospective analysis from a high-volume centre” is a retrospective study that evaluates early hemorragic complications after holmium laser enucleation of the prostate in patients undergoing antithrombotic therapy. There are few aspects that I like to mention:

Comments 1: The retrospective nature or the study limits the ability to establish causality and may introduce selection bias. This design can lead to confounding variables that are not adequately controlled.

Response 1: Thanks for this point. As well known, retrospective studies are susceptible to selection bias, and this represents the major limitations of our study, as acknowledged in Discussion section. However, we underline that available evidences come exclusively from retrospective studies, probably due to difficulty in carrying out RCTs in this specific setting: in fact, discontinuation of AP/AC therapy is not always possible because of a high risk of serious thromboembolic complications, and randomly select patients to continue or stop the therapy may not be feasible for ethical reasons.

Comments 2: Although the study claims to be among the larger reports regarding NOACs (New Oral Anticoagulants) before HoLEP, the overall sample size of 338 may still be inadequate to draw definitive conclusions, particularly when stratifying outcomes between specific antithrombotic medications (e.g., NOACs vs. warfarin).

Response 2: The point is well taken. Up to date, only Becker et al. in 2019 [doi: 10.1089/end.2018.0693] reported a larger case series of patients on NOACs therapy undergoing HoLEP (94 vs 47 patients in the present study). We agree with the reviewer that these numbers are still low for stratifying results according to the specific drug, and this aspect is highlighted among limitations. It is worth remembering that NOACs have been introduced in clinical practice “only” in the last decade. Given the advantages of those drugs over warfarin, nowadays NOACs can be considered the gold standard in AC therapy: indeed, this “therapeutic shift” is evident in our results, with 94% of patients under NOACs. Nevertheless, it would be necessary to have larger numbers, perhaps deriving from multicentric studies, to better identify the safety profile of different NOACs in HoLEP patients.

 Comments 3: Lack of Distinction Between Antithrombotic Drugs: The study groups patients into broad categories (no therapy, AP therapy, AC therapy) without sufficiently analyzing the nuanced effects of different drugs within those categories. For instance, the outcomes of various NOACs or the effects of specific AP therapies were not evaluated in detail, which could affect the generalizability of the findings.

Response 3: Thanks for this point. Accordingly with previous studies cited in Discussion section, we stratified study groups into broad categories. As already mentioned in response #2, 47 patients (for AC group) and 76 patients (for AP group) are a limited number to carry out subgroup analyses according to specific drug. We thought reasonable to stratify patients as stated above for two reasons: 1) almost all patients on AC therapy were intaking NOACs, which share similar mechanism of action, indications, perioperative management, etc.; 2) as stated in latest EAU guidelines on Thromboprophylaxis [https://uroweb.org/eau-guidelines/discontinued-topics/thromboprophylaxis], recommendations on AP agents perioperative management are based on the results of a RCT comparing aspirin to placebo [doi: 10.1056/NEJMoa1401105], while there is a lack of placebo-controlled trials for other AP drugs; however, given similar antithrombotic and bleeding profiles, the indirect evidence is considered valid for all AP drugs. From a practical standpoint, given the limitations of our study, we considered this stratification sufficient for evaluate early hemorragic complications after HoLEP in this setting.

Comments 4: The inclusion/exclusion criteria may introduce bias. For example, patients with a history of prostate or bladder cancer, which may relate to different risk profiles and complications, were excluded, potentially skewing results in favor of a more homogeneous patient population.

Response 4: Thanks for this point. Actually, we excluded all patients potentially representing confounders, since they could be characterized by an intrinsic higher risk of post-operative bleeding. It is our opinion that a more homogeneous population can ease the identification of the actual effect of AP/AC drugs.

Comments 5: Postoperative Outcomes Limited to Uroflowmetry: The evaluation of postoperative functional outcomes is limited solely to uroflowmetry parameters. This method does not capture holistic patient satisfaction or other relevant health-related quality-of-life indicators that could provide better insight into the procedure's overall efficacy.

Response 5: Thanks for raising this point. Unfortunately, post-operative IPSS and QoL scores were available for a very limited number of patients, making any statistical analysis on patient-reported outcomes unreliable.  As stated among limitations, this was due to the retrospective nature of the study. Nonetheless, since this was not the main objective of our study, given the lack of differences among groups in post-operative maximum flow rates, it is plausible to state that antithrombotic therapy does not influence functional outcomes.

Comments 6: Inconsistent Preoperative Management: The management of antithrombotic therapy showed variability in application, particularly regarding bridging therapy. The inconsistency in management protocols across patients (e.g., varying timing for stopping and restarting therapies) could confound outcomes and complicate the interpretation of bleeding risk.

Response 6: Thanks for the comment. Probably the greatest interpretative difficulties concern perioperative management of warfarin, since bridging with LMWHs can continue for a variable period (up to 7 days), potentially representing a bias. On the other hand, for all the other agents (AP and NOACs) perioperative management is well codified, as stated by EAU guidelines (see above). However, since only 3 out of 338 patients received warfarin in out study cohort, we believe that the risk of bias is negligible.

Comments 7: Incomplete Pharmacological Anamnesis: The absence of a complete pharmacological history limits an understanding of potential drug interactions and their effects on bleeding risk, which has been highlighted as a critical area of concern for patients undergoing surgery while on antithrombotic therapy.

Response 7: This is a very nice comment. A complete pharmacological anamnesis was not available, as stated among limitations of the study. As correctly highlighted by the reviewer, polytherapy in patients intaking antithrombotic therapy may lead to drug interactions which could potentially translate into an higher risk of post-operative bleeding. We underline that multivariate analysis of the logistic regression model showed that only AC therapy (not AP therapy) is an independent predictor of early hemorragic complications. Since almost all patients on AC therapy were intaking NOACs, is paramount to focus our attention on drug-drug interactions for these drugs. A recent review by Foerster et al. [doi: 10.1007/s40262-020-00879-x] concluded that many co-medications can interact with NOACs and modify their exposure; however, the effect of multiple drugs on NOACs exposure in patients with normal renal function, and especially the impact of drug-drug interactions, has not been fully elucidated; for the same reason, NOACs dose adjustments based on concentration measurements cannot be recommended because evidence-based data are missing. These considerations were added in Discussion section (page 10, lines 278-282).

Comments 8: No Randomized Control: The lack of a control group that includes a randomized setup could bias results. It would be beneficial to compare HoLEP outcomes in a controlled setup among patients with similar health backgrounds but differing in anticoagulant therapy type.

Response 8: We thank the reviewer for the comment. As already stated in response #1, carrying out RCTs in this specific setting could be difficult. We strongly believe that future research should aim for large prospective studies to better define the safety profile of different antithrombotic drugs in HoLEP patients.

Comments 9: Temporal Considerations: The study acknowledges that differences in outcomes appear postoperatively, around the second day, coinciding with the resumption of antithrombotic therapy. This temporal aspect may suggest that some outcomes could be mediated by factors linked to the timing of drug actions, which were not thoroughly examined.

Response 9: Thanks for the raising point. This aspect was further examined in Discussion section (page 9, lines 258-260).

Overall, while the study contributes valuable insights into the interaction between HoLEP and antithrombotic therapy, significant methodological limitations and biases must be addressed before considering these findings as definitive. Future research should aim for prospective, randomized controlled trials with larger and more diverse populations, comprehensive pharmacological assessments, and a broader range of outcome measures to enhance the reliability and applicability of the results. This could ultimately establish clearer guidelines for managing patients on antithrombotic therapy undergoing HoLEP, thereby improving clinical outcomes and patient care.

Reviewer 3 Report

Comments and Suggestions for Authors

The present retrospective study shows more data about the complications of HOLEP in patients under antithrombotic therapy. The cohort is small but the percentage of patients under this therapy is significant taking into account what have been published previously.

They also review the previous data in this setting, which is mostly retrospective.

There are some concerns about the cohort:

-        Introduction could further contextualize the problem

-        As mention, why the percentage of antithrombotic therapy is so high in this cohort in comparison with the cohorts previously published. Is there any specific putative different indications?

-        The descriptive data could be better represented with also ranges to better show the cohorts characteristics, but also the results.

-        Mean LOS and catheter time remain statistically significant but, are these results clinically relevant? Could you further explain your protocol in terms of catheter removal and patient discharge? Could you further debate about the clinically relevance of your numbers?

Author Response

The present retrospective study shows more data about the complications of HOLEP in patients under antithrombotic therapy. The cohort is small but the percentage of patients under this therapy is significant taking into account what have been published previously.

They also review the previous data in this setting, which is mostly retrospective.

There are some concerns about the cohort:

Comments 1: Introduction could further contextualize the problem

Response 1: Thanks for this suggestion. The Introduction section was expanded as suggested (page 3, lines 51-62), and specific references were added as well.

Comments 2: As mention, why the percentage of antithrombotic therapy is so high in this cohort in comparison with the cohorts previously published. Is there any specific putative different indications?

Response 2: Thanks for this point. The higher percentage of antithrombotic therapy is probably related to the fact that Aurelia Hospital is classified as Second Degree Department of Emergency: in Italian healthcare system, these structures represent referral centres for Interventional Cardiology and Cardiovascular Surgery. Consequently, the percentage reflects this aspect, with more patients under AP/AC therapies referring to Urology Unit for BPH surgery.

Comments 3: The descriptive data could be better represented with also ranges to better show the cohorts characteristics, but also the results.

Response 3: Thanks for this suggestion. Data were modified accordingly in all tables.

Comments 4: Mean LOS and catheter time remain statistically significant but, are these results clinically relevant? Could you further explain your protocol in terms of catheter removal and patient discharge? Could you further debate about the clinically relevance of your numbers?

Response 4: Thanks for this point. In order to get reimbursement after a procedure for BPO, the regional public healthcare system to which Aurelia Hospital refers requires at least 2-night stay, which inevitably leads us to discharge our patients not before the 2nd post-operative day. In our experience, a post-operative protocol that guarantees the best compromise in terms of patients’ comfort, catheter-related adverse events (i.e. dysuria, hematuria, transient urinary retention) and administrative management is made up of: 1) stopping bladder irrigation on the 1st post-operative day; 2) catheter removal on the 2nd post-operative day; and 3) discharge after spontaneous micturition. The results of our study are consistent with the features of this protocol. We agree with the reviewer that, although statistically significant, probably the differences between groups are less clinically meaningful, except for costs related to hospital re-admissions and transfusions.

Round 2

Reviewer 3 Report

Comments and Suggestions for Authors

Thank you for your responses.

Please further revise and include interquartile range in table 2 for quantitavie varaibles (some of them have not been changed).

Also it would be nice to introduce this sentence in methodology as part of the protocol:

1) stopping bladder irrigation on the 1st post-operative day; 2) catheter removal on the 2nd post-operative day; and 3) discharge after spontaneous micturition.

Also in discussion section the reason about the high percentage of patient under these drug could be explain in order to extrapolate results to other centers

Mean catheter day should also be shown as qualitative showing the global percentage of patients that could not follow this protocol because haematuria.

Author Response

Thank you for your responses.

Please further revise and include interquartile range in table 2 for quantitavie varaibles (some of them have not been changed).

Response: Thanks for this suggestion. The variables “catheter time” and “length of hospital stay” were reported as mean ± standard deviation in order to better underline differences between groups. However, as suggested by the reviewer, these variables are now reported as median and IQR.

Also it would be nice to introduce this sentence in methodology as part of the protocol:

1) stopping bladder irrigation on the 1st post-operative day; 2) catheter removal on the 2nd post-operative day; and 3) discharge after spontaneous micturition.

Response: Thanks for this point. The protocol was specified in Materials and Methods section (page 5, lines 113-115).

Also in discussion section the reason about the high percentage of patient under these drug could be explain in order to extrapolate results to other centers

Response: Thanks for this point. This issue was addressed in Discussion section (page 10, lines 280-283).                                                   

Mean catheter day should also be shown as qualitative showing the global percentage of patients that could not follow this protocol because haematuria.

Response: Thanks for raising this point. In Table 2 was added the categorical variable “deviations from standard protocol” (i.e. percentage of patients that could not follow the standard post-operative protocol because of macrohematuria). Chi-square test showed a statistically significant difference for this variable between groups (p = 0.001), as stated both in Table 2 and Results discussion (page 6, lines 150-152).